# Competition in a Household Energy Conservation Game

Jan Dirk Fijnheer *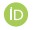, Herre van Oostendorp, Geert-Jan Giezeman and Remco C. Veltkamp

Department of Information & Computing Sciences, Utrecht University, Princetonplein 5,
3584 CC Utrecht, The Netherlands; h.vanoostendorp@uu.nl (H.v.O.); g.j.giezeman@uu.nl (G.-J.G.);
r.c.veltkamp@uu.nl (R.C.V.)
* Correspondence: J.D.L.Fijnheer@uu.nl

**Abstract:** This paper presents the results of a game study, comparing Powersaver Game including a competition feature versus the same game excluding a competition feature with respect to energy conservation in the household. In a pretest–posttest design, we tested whether change in attitude, knowledge and behavior with respect to energy conservation in the household was different for participants playing Powersaver Game with or without competition. All energy conservation activities that the application provides (e.g., washing clothes at low temperatures) take place in the real world and feedback is based on real-time energy consumption. This so-called reality-enhanced game approach aims to optimize the transfer between the game world and the real world. Household energy consumption changed significantly and positively in the long term due to competition. A significant difference of 8% in energy consumption between both conditions after the intervention was detected. Besides energy conservation, no further differences were detected between conditions. The chain of events, that an increase in knowledge leads to attitude change, which in turn results in behavior change in the long term is confirmed by means of a path analysis. We conclude that Powersaver Game is effective in the transfer of energy conservation knowledge, which leads to energy saving behavior in the long term while competition additionally contributes to more change in behavior.

**Keywords:** gamification; energy conservation; user studies; reality enhanced games; attitude change; behavior change; competition



## 1. Introduction

In the field of persuasive games, a blend of real and virtual game elements is used to create gaming experiences. Real-world games are augmented with computing functionality, or as with gamification, virtual computer entertainment is integrated into the real world [1]. A persuasive application that aims to stimulate energy conservation is more effective when game mechanics like missions, quizzes, narrative, competition and rewards are implemented. Additionally, besides game mechanics, the inclusion of real-world activities by using 'reality enhanced games' principles in a persuasive application is an outstanding effective means to change people's energy conservation behavior [2–4]. In these 'reality-enhanced games', energy reduction is digitally obtained from the smart energy meter and is fed into the digital game in various compelling ways. For example, energy reduction by lowering the thermostat will result in positive feedback (e.g., competition ranking, badges, etc.) and progression of the game's storyline. Powersaver Game is used as a tool in a larger research project that examines the influence of playing in the real world on attitudes towards energy conservation, and on energy conservation behavior in the long term. The focus is specifically on energy consumption in households by means of electricity and gas usage. The aim is to contribute to the stimulation of individual sustainable behavior by studying how gamification can be a positive incentive for people to change their behavior regarding energy use at home by attitude change. It also aims to study whether transfer from game play to real-life behavior has a long-term character. Furthermore, it is conducted

over a longer period of time, measures changes in knowledge, attitude and behavior also after delay, and includes an adequate control condition [5].

In previous research, a 'media comparison' study [6], we concluded that there are differences in learning the same content of a persuasive energy conservation game, developed by using an iterative user-centered game design methodology, compared to an energy dashboard control condition. Furthermore, we concluded that energy consumption changed significantly in the long term. A persuasive game that includes reality by using reality enhanced games principles is, thus, effective in learning people to save energy in the household and to actually do that for the long term, while an energy dashboard does not change that behavior at all [3,7,8]. A recent publication by PBL Netherlands Environmental Assessment Agency, the national institute for strategic policy analysis in this field, also confirms the limited effects of energy dashboards [9]. These results are in line with the main conclusions of a meta-analysis of Clark et al. [10]. Their first conclusion is that games enhance learning relative to nongame conditions, and their second conclusion is that games that incorporate reality enhance learning more than standalone games.

After the previous 'media comparison' study, our research in persuasive games focuses on specific game features that contribute to the game's persuasiveness to promote lasting changes in knowledge, attitude and behavior regarding sustainable energy use of households [11]. There is a need for conducting more empirical research on how different game elements can enhance learning outcomes [12–16]. For that purpose, we have, in the next phase of research, applied a 'value added' approach. Studies using this approach focus on the question of which features of a game promote learning. Conditions including preselected features are compared to a condition with a base version of the game [6]. We examine the effects of the persuasive feature social interaction by means of competition on participants' knowledge, attitude and behavior with respect to sustainable energy consumption with Powersaver Game. In general, competition is a fundamental game feature [17] that should enhance motivation and stimulate cooperation and learning. However, research provides limited empirical evidence of effects when intervention periods are long and competition is applied in less-structured domains [12,18,19].

Our focus is comparing a group of households that play Powersaver Game with the competition feature (intervention condition) and a similar group of households that also play Powersaver Game but without the competition feature (control condition). We hypothesize that knowledge, attitude and energy conservation of participants playing the game with the competition feature will increase more than that of participants in the control condition because we expect that social comparison between households by means of competition will stimulate goal orientation with respect to competitors/other households, and therefore enhance collaboration within households [12,18,19]. Therefore, they will make more effort to accomplish missions that take place in the real world (e.g., washing clothes at low temperatures and taking shorter showers) than households in the non-competition condition, and therefore probably attain better results.

This paper contributes to empirical research that focuses on the long-term effects of the competition feature on knowledge, attitude and behavior, using a reality enhanced games approach to stimulate energy conservation of households.

In the next section the theoretical background is discussed, where we pay attention to gamification and the game features competition and collaboration. In the third section the research design is presented, where we pay attention to game design and technical realization. In the fourth section, the outcomes of the empirical study are discussed. Finally, in the fifth section, we draw conclusions and discuss how we will continue our research to bring the research field on energy reduction games theoretically, but also practically, further.

## 2. Theoretical Background

### 2.1. Gamification

Deterding et al. [20] stated that gamification by incorporation of game features can be a valuable strategy for making non-game products, services, or applications, more

motivating, and/or engaging to the user. From this broad view, a taxonomy of five approaches of gamification can be formulated based on complexity, main characteristics and game features. This taxonomy of approaches gives more differentiation, depth and scientific relevance to the diverse appearances of gamification. As presented in Table 1, the first is the simplest form where a playful persuasive element in a creative design stimulates simple behavior [21]. In the second approach, Feedback systems, actual behavior is measured and results are presented to the user [22]. In the third approach, Learning systems, a learning loop is created when first instructions for certain behavior are given after which later feedback is presented [23]. From this approach on, applications are considered to belong to Digital Game Based Learning (DGBL) [24,25]. The fourth approach consists of complex standalone serious (simulations) games with several, more complex, game mechanics (e.g., storylines and competition) included [26]. An important aspect of serious (simulations) games is that users normally gain implicit knowledge (improved performance during the game), but this gain does not always translate into a gain in explicit knowledge (e.g., improved performance on knowledge tasks after the game or transfer tasks) [27]. This problem is mitigated by the last and most complex approach of gamification, namely 'Reality Enhanced Games'. The aim of this approach is to optimize the transfer between the game world and the real world. When the transfer is optimized, the game is expected to be more effective in the change of behavior and attitude [28]. In this approach user's real-world activities feed a digital serious game or gamified application. Players are immersed in real-life situations that are generated by user interaction with a virtual environment [3,4].

**Table 1.** Taxonomy of gamification approaches.

| Approach | Complexity | Main Characteristic and Game Features | Example |
|---|---|---|---|
| Playful persuasive element [21] | Very low | Stimulate behavior by creative design | Piano stairs to encourage to take the staircase |
| Feedback systems [22] | Low | Playful feedback | Pedometer to increase physical activity |
| Learning systems [23] | Medium | Learning loop by means of playful instruction and feedback | Mental health apps |
| Serious (simulations) games [26] | High | Standalone game | Military simulations |
| Reality enhanced games [3,4] | Very high | Gameplay is real world process | Household energy conservation |

While it has been technically possible to implement real world processes in a game design for more than a decade, it is still an emerging principle in gamification research. Especially when it comes to the energy conservation of households. Research has shown that the integration of serious games into real-life can have positive effects on attitude and behavior [2–4]. When people are highly engaged, they are apt to adopt the attitude that is promoted in the application [29]. This can lead to a higher awareness of relevant factors involved in, for instance, energy conservation. In effect, attitude may positively change, and subsequently, trigger a change in energy-saving behavior in the long term. As presented in Figure 1, the assumed chain of events that higher awareness (an increase of more accessible knowledge) leads to attitude change, which in turn leads to behavior change, is what more complex approaches of gamification try to accomplish [2,3,5,30,31].

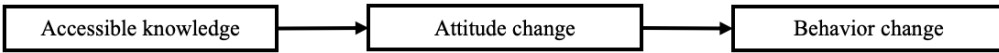

**Figure 1.** Assumed chain of events in behavior change.

## 2.2. Competition and Collaboration in Digital Game Based Learning

The game features competition and collaboration are closely related because they are both influential in creating interaction between players [19] and performance-awareness [18],

and as a result, both have positive effects on motivation, attitudes and behavior [12,18,32]. However, research provides limited explanation of how exactly these features influence learning, and there is a lack of guidelines for game designers [18]. Although in many DGBL studies no difference was found between competition and cooperation in terms of learning achievement [33–35] found that the effectiveness of collaboration was negatively affected by competition specifically for low achieving students. On the other hand, Chen et al. [36] suggest that collaboration within groups will increase when groups are in competition because players are working toward a common goal. Some studies found differences in motivation in the short and long term and by the complexity of tasks that have to be performed. Dindar et al. [37] and Katz-Navon and Erez [38] both suggest that a DGBL-setting collaboration invokes higher task effort in long-lasting learning activities compared with competition, and Chen [39] reports that collaboration increases reflective thinking and effective problem-solving. Therefore, collaboration seems to be an effective feature for higher and complex learning tasks that takes a longer time period to carry out. Also, Marker and Staiano [40] report a similar difference between both features. They found that the competition feature in a fitness game typically stimulates motivation and behavior in a short time period, such as in a game session. While the collaboration feature stimulates motivation for a longer period, such as continuing playing several game sessions over time.

### 2.2.1. Collaboration

Collaboration in DGBL involves problem-solving and constructing knowledge together as a team. This takes mutual engagement and coordinated efforts [18,41]. Participants extend their knowledge and have to make it explicit to others. This makes interaction in a collaborative setting highly engaging [42]. Its effectiveness depends on the quality of dialogues in acquiring knowledge from each other by questioning, answering, and discussing [18,41]. These discussions, about conflicting information, lead to opportunities for reflection on the offered content and present knowledge [13], and thus stimulate information exchange and constructive communication [37]. To be more specific: When asking questions, participants outline what they know and/or identify what they need to know, which helps to become aware of their knowledge and to generate knowledge. In this way both learning processes and outcomes benefit from collaboration [42] and, through these social interactions, also positive motivational experiences are facilitated and feelings of relatedness generated [43]. Furthermore, when team members face emotional challenges such as frustration or demotivation collaboration stimulates mutual cognitive, motivational and emotional support [37,44].

### 2.2.2. Competition

Competition is a rudimentary gaming element [17] that stimulates goal orientation with respect to competitors [19]. Therefore clearly-defined goals, fair rules and social comparison opportunities enhance motivation [18,37,45] and as a consequence stimulate learning [12,19]. Competition also positively influences excitement, perceived challenge, effort, efficacy, game frequency and enhances collaboration within teams [12,18,19]. Competition seems to be more or less similar beneficial for different types of users [46], however excessive competitive activities may cause negative influences on learning such as anxiety, damage of relationships, impeding performance on tasks, diminishing empowerment and irresponsibility for learning [47]. Although most authors report positive effects of competition, there is ambiguity on how competition influences the learning outcomes [12,18,19]. Chen et al. [12] stated that competition is effective in well-structured domains like math, language learning, and science. But still, it is unclear what its effectiveness is in less-structured domains such as social science. Chen et al. [12] also found different effects of competition in different DGBL game types. Effects of competition in DGBL were significant for Role-playing games, Simulation, Puzzle and Strategy games, but not for Action games.

Van der Cruysse et al. [45] describe three approaches of competition in DGBL. The first is individual and team-based competition approach. In the individual competition

approach, each individual is an autonomous player, and in the team-based competition approach, teams of players compete. Chen [39] found that the team-based competition approach is more effective in learning outcomes and problem-solving, and there is less anxiety than in individual competition approaches. Furthermore, van der Cruysse et al. [45] found that team-based competitive approaches are especially effective in making instructional materials more enjoyable and engaging. The second approach of competition is anonymity of opponent(s). Players know or do not know their opponents. Cagiltay et al. [19] found that competition approaches that allowed players to see each other's scores, ranking and names enhanced learning and motivation. The third and last approach of competition involves reality-based versus computer-based opponents. While real opponents can be more motivating, computer-based competition addresses several disadvantages of real competition [48]. Kristan et al. [49] recommend computer-based competition in which difficulty and chance of failure are adapted to the individual user. Adaptation provides the user with the right amount of challenge to maximize motivation and as a consequence stimulates learning. Chen and Chang [48] report a significantly better learning performance and time effort with virtual competition. Van der Cruysse et al. [45] conclude that the effectiveness of the three approaches of competition depends on the context where the game is applied, since competition functions differently (e.g., in learning outcomes, behavior and motivation) in different situations (e.g., type of game and opponent (oneself, other(s) or computer)).

*2.3. Related Work on Energy Conservation Games*

The game we have developed for our research project, Powersaver Game, focuses on energy use and conservation at home, specifically where personal behavior is involved. Similar serious games that also focus on energy use and conservation were chosen based on six criteria. To be selected, a game: (1) has to raise awareness concerning energy conservation at home, (2) transfers information about energy consumption so that players acquire more knowledge, (3) influences players to change their behavior concerning energy consumption in real life, (4) integrates behavior in real life into the game by monitoring energy consumption in real life and using this information in the game progression, (5) is played over a relatively long period of time and has several sessions and (6) has a compelling and complex storyline that is able to engage players. Searches were performed in scientific databases to assess games that had been used as a research instrument with partial similarities to our research. As presented in Table 2 eight games appeared to be relevant. Most studies regarding energy conservation of households that evaluate the effect of gamification features in general and competition specifically have shortcomings and the reported results are sometimes ambiguous [5,50,51]. However, the empirical effects that are explicitly reported, in changing knowledge, attitude and/or behavior are positive. Furthermore, we also found that five games reported that collaboration is an effective core game design feature, and three games reported that competition is an effective core game design feature. Two games reported both features are effective core game design features.

**Table 2.** Comparative analysis energy games.

| Game/Features | Collaboration | Competition |
|---|---|---|
| 1 The Power House [52] | Not mentioned | Not mentioned |
| 2 Power Agent [53,54] | Positive effect | Positive effect |
| 3 EcoIsland [55] | Positive effect | Positive effect |
| 4 Power Explorer [56,57] | Positive effect | Not mentioned |
| 5 Agents Against Power Waste [58] | Positive effect | Not mentioned |
| 6 EnergyLife [59,60] | Not mentioned | Positive effect |
| 7 Power House [61,62] | Positive effect | Not mentioned |
| 8 Energy Chickens [63] | Not mentioned | Not mentioned |

We reported on these studies before, see Fijnheer and van Oostendorp [2] and Fijnheer et al. [50]. Unfortunately, none of these studies use a 'value added' approach or included a control condition, and had several shortcomings such as: (1) a short intervention time, (2) no real consumption measurements are used, (3) poor implementation of gamification features, (4) limited number of variables are measured, (6) the lack of pre-measurements and post-measurements [2,50].

In the next section, the research methodology is discussed, where we pay special attention to game design and technical realization.

## 3. Research Methodology

### 3.1. Research Question

A value added approach examines the instructional effectiveness of game features within DGBL applications, answering the research question: "Which game features of a DGBL application exactly promote lasting changes in knowledge, attitude and behavior?" In a value added experiment, the primary independent variable is the presence or absence of an instructional feature, in this case, the competition feature. The main research question concerns whether people learn better when an instructional game feature is added to a game [6]. Inspired by this approach, our focus is to examine the effects of the feature social interaction, by means of competition, on participants' knowledge, attitude and behavior with respect to sustainable energy consumption with Powersaver Game compared to a base version of Powersaver Game.

Powersaver Game is a reality-enhanced game where several gamification mechanics, such as a narrative, missions, quizzes, avatars, competition, gameplay, feedback and reward systems, are implemented and can be expected to stimulate energy conservation behavior of households. Before the intervention starts the competition feature can easily be turned on or off to create an intervention condition and control condition of families. In both conditions, everyday families receive the same information about energy conservation measures and receive feedback. Each mission revolves around a specific theme, such as washing clothes, cooking or media use. We measure knowledge transfer, i.e., learning results of participants, but also their attitude and behavior, i.e., energy consumption.

### 3.2. Participants

In this study, 18 households including 31 participants older than 12 years participated on a voluntary basis in this experiment. The households consist of 11 families with children, 4 families without children and 3 single-person households. No households dropped out during the intervention.

### 3.3. Design

Powersaver Game is a web-based application that runs on an internet browser. The navigation by the player is done by 'point and click' in the browser. The game is played in households, involving all persons living in that household. The game has similarities with Eco-feedback -, Multiplayer -, Roleplaying- and Point & Click Adventure games [64] and has been designed in an iterative design process, in which several design steps have been made [50]. In order to obtain data on the energy consumption of households in the game, the Dutch grid operator establishes a real-time connection between the household energy meter and the game server. On a daily basis, data on energy consumption is sent to a database of a server at Utrecht University. See hereafter Section 3.3.3 Technical architecture.

The central characters of Powersaver Game are avatars that represent family members of the real household (see Figure 2). When participants start the game for the first time, an introduction to the story is presented. A game with a storyline can be engaging because it can stimulate our emotions [65,66]. The story starts at the moment that a family arrives at a dilapidated country house. The story is presented through text blocks and explains that a professor had caused a failed experiment in this country house (see Figure 3). Then the family enters the house and stands in the main hall with several doors (see Figure 4).

Behind each door a room, e.g., kitchen, bedroom or bathroom, is situated in which a game character in the form of a confused electrical device has been located. During the mission sessions, the family is accompanied by a cat named Kyoto, the professor's former pet.

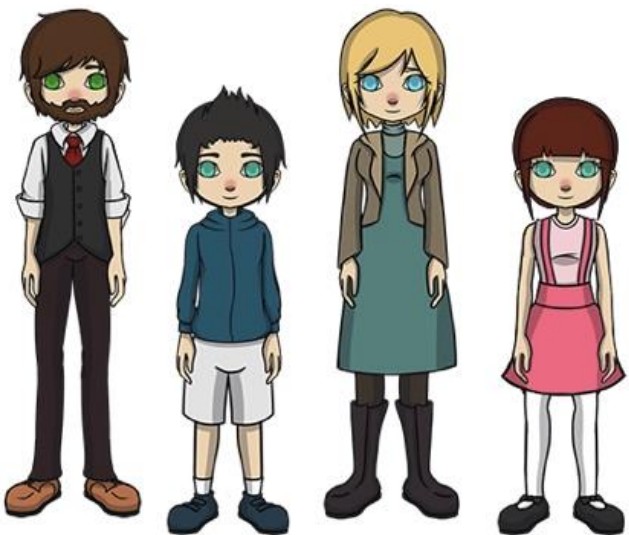

**Figure 2.** Avatars.

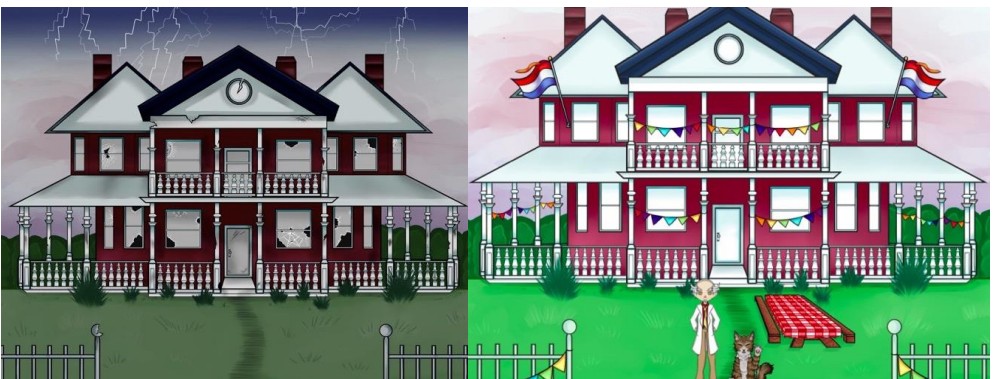

**Figure 3.** House of the professor; bad begin-state (on the **left**) and normal end-state (on the **right**).

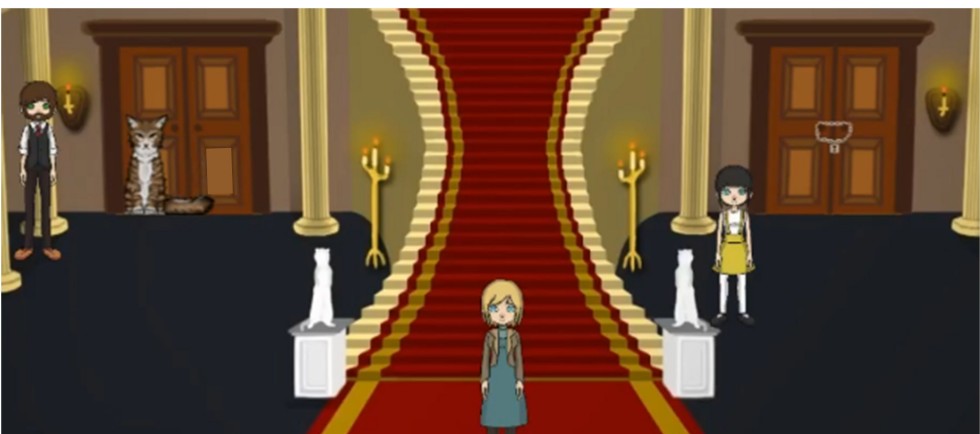

**Figure 4.** Part of the main hall.

Almost every day a mission session is presented to the family. When they open the game, they are asked to enter a preselected room in the main hall. When they click to open the door, a quiz appears first. This quiz contains multiple-choice questions about

energy conservation measures related to the room behind the door. For example, questions about cooking when the kitchen is behind the door. The quiz questions will assess players' knowledge for the missions that are occurring in that specific room. When the quiz has ended the family enters the room, and a character in the form of a device that is in a confused state is shown (see Figure 5, on the left). The family has to accomplish missions in their household, which involve energy conservation knowledge, to help the device to return to a normal state. All missions, including fifty energy conservation activities in all, such as washing clothes at low temperatures, taking shorter showers, turning down the thermostat and using eco-programs on devices, take place in the real world. This represents our previously described reality-enhanced game principle. The missions are presented in an active writing style because it is directly related to the urgent situation in the narrative. If households end missions and start new ones in the given time, the game is finished in at least 3 weeks. Depending on the complexity, it takes about a day to complete a mission. There are approximately thirteen missions, eight quizzes and a final-battle/scene in Powersaver Game. The end situation of a mission is reached when the device is brought out of its confused state (see Figure 5, on the right). Then a new mission can be opened which is located behind another preselected door. The entire game is ended when all devices from all rooms in the country house and the professor are brought out of their confused state (see Figure 3, on the right). A separate page in Powersaver Game provides feedback on energy consumption and energy conservation during gameplay (see Figure 6). Feedback on energy conservation is based on the average energy consumption in the 21 days before the intervention started. The results of the quizzes and achievements of completed missions, both expressed in badges, are also presented on a separate page (see Figure 7).

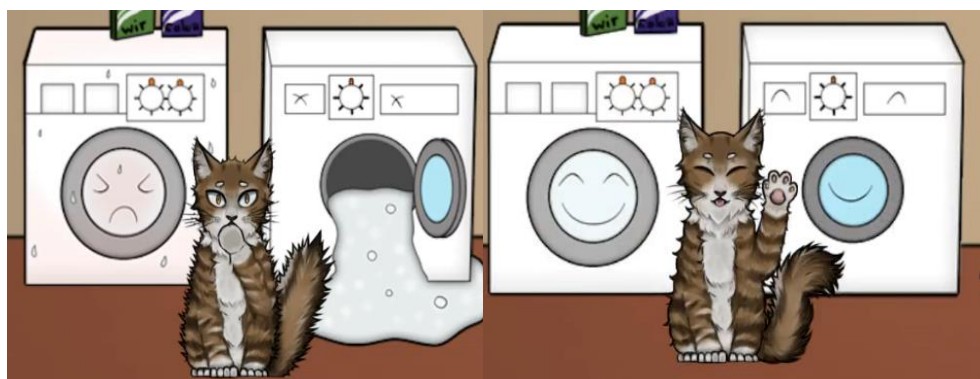

**Figure 5.** Laundry scene; bad state (on the **left**) and normal state (on the **right**).

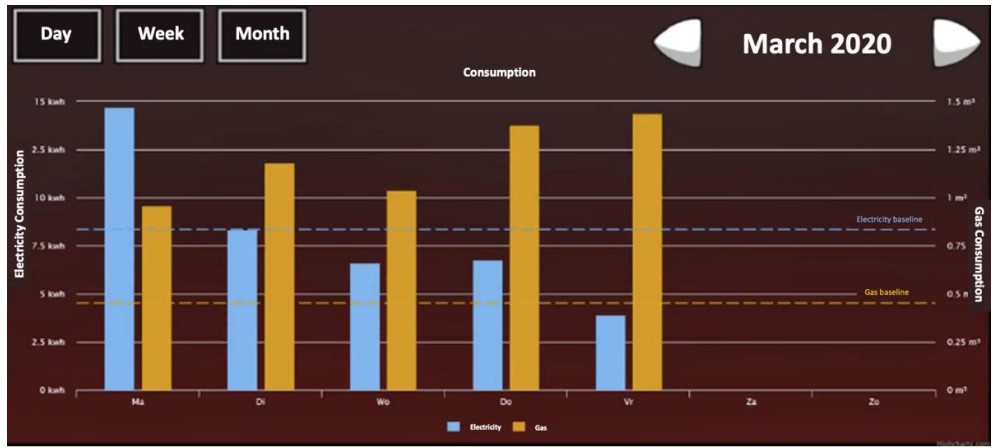

**Figure 6.** Consumption chart.

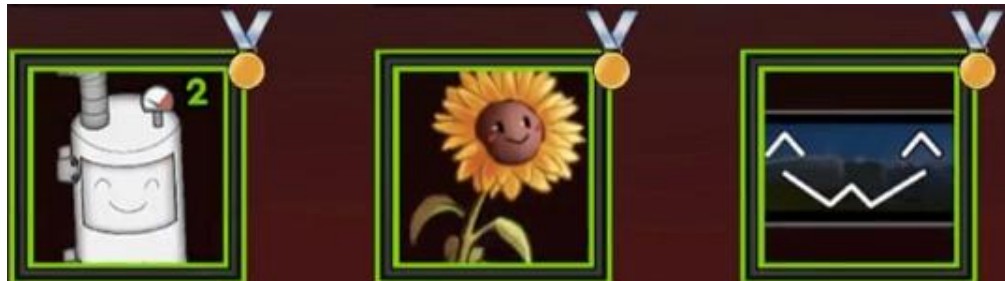

**Figure 7.** Badges.

### 3.3.1. Competition Feature Design

Powersaver Game has a team-based approach of competition, because all persons in the household are involved and form a team. It is expected that this approach stimulates collaboration, and as a consequence makes energy conservation measures to be more enjoyable and engaging. Furthermore, through this team-based approach, the players are aware that for success, collaboration is necessary [67].

Competition features should be designed in a way that the experience of uncertainty in winning always remains to the ending [18]. Therefore, each household in the intervention condition is in competition with 9 virtual (computer-based) households but assumes to play against 9 real households. Competition is simulated to stimulate households to achieve high scores. This way positive influences of social comparison opportunities are stimulated and negative influences prevented like frustration, discouragement and potentially dropping out of less-able players who always lose while more-able players always win [37]. Negative feedback was not part of the manipulation. The assumption to compete with real households, which are actually virtual households, should enhance learning and motivation. Players have the impression to be assigned to equal opponents with similar abilities [68]. Beside these advantages, it was technically not feasible to implement a real and fair competition.

The competition feature is displayed as a simulated leaderboard where, through displays of rank, comparisons with other virtual households are presented (see Figure 8). Nebel et al. [69] describe this design as 'artificial social competition' where opponents offer humanlike features, by means of real-looking scores and family names, without actually being human but simulated by a computer algorithm. The energy conservation data of virtual households follow a logical pattern based on the real-time energy conservation results of the real household (see Figure 9). The black line represents the real household of the participant competing against nine other households. The scores of the top 4 ranking households, including the real household, are close to each other, and therefore should stimulate desired behavior.

### 3.3.2. Control Condition

For our approach in this study, families used Powersaver Game without the competition feature in the control condition. The form, timing and content of the information the control condition receives are similar as in the intervention condition, but with the exclusion of the game feature competition. The design of both versions of Powersaver Game is identical except the option 'Rankings'—right above in Figure 8, which represents the competition feature—is absent in the control condition.

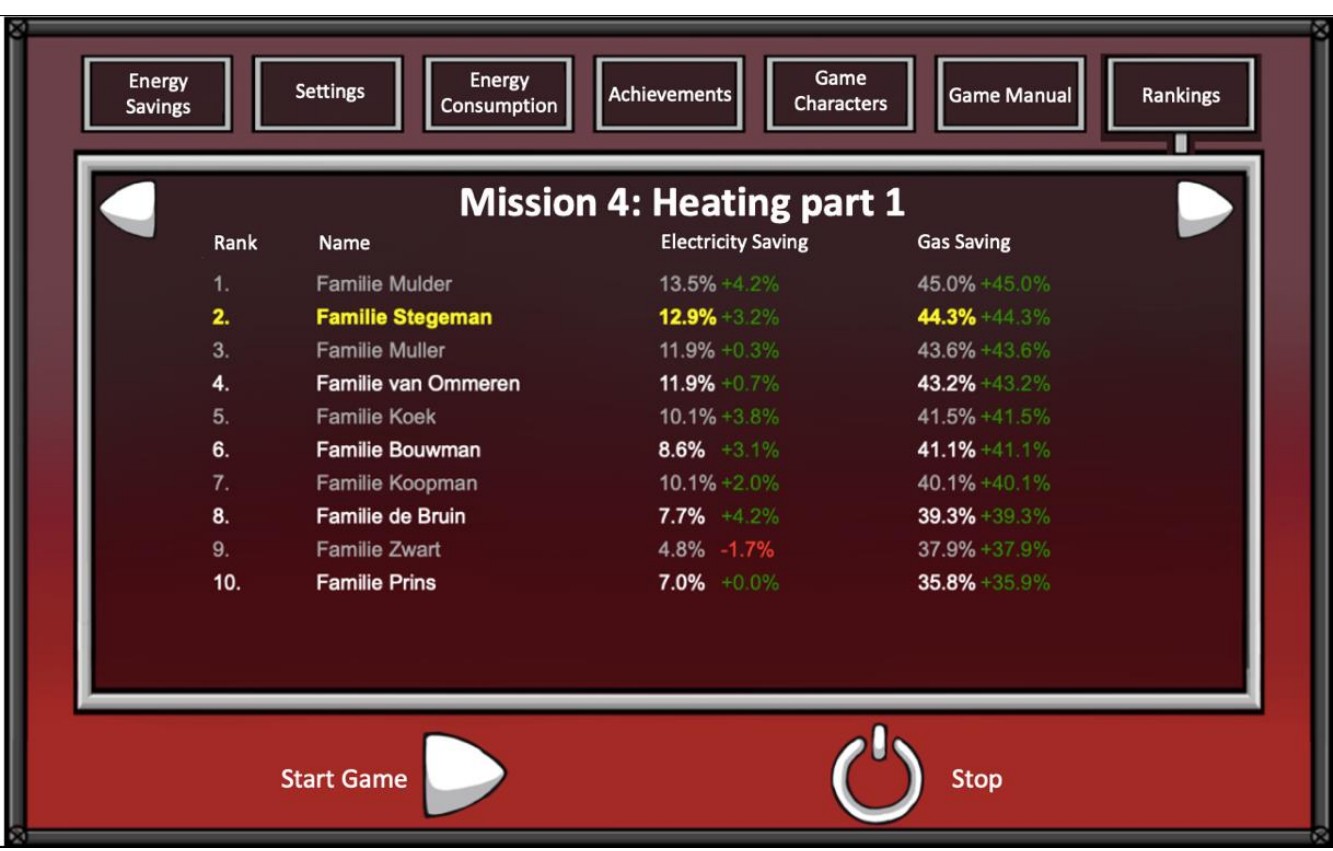

**Figure 8.** Competition feature Powersaver Game.

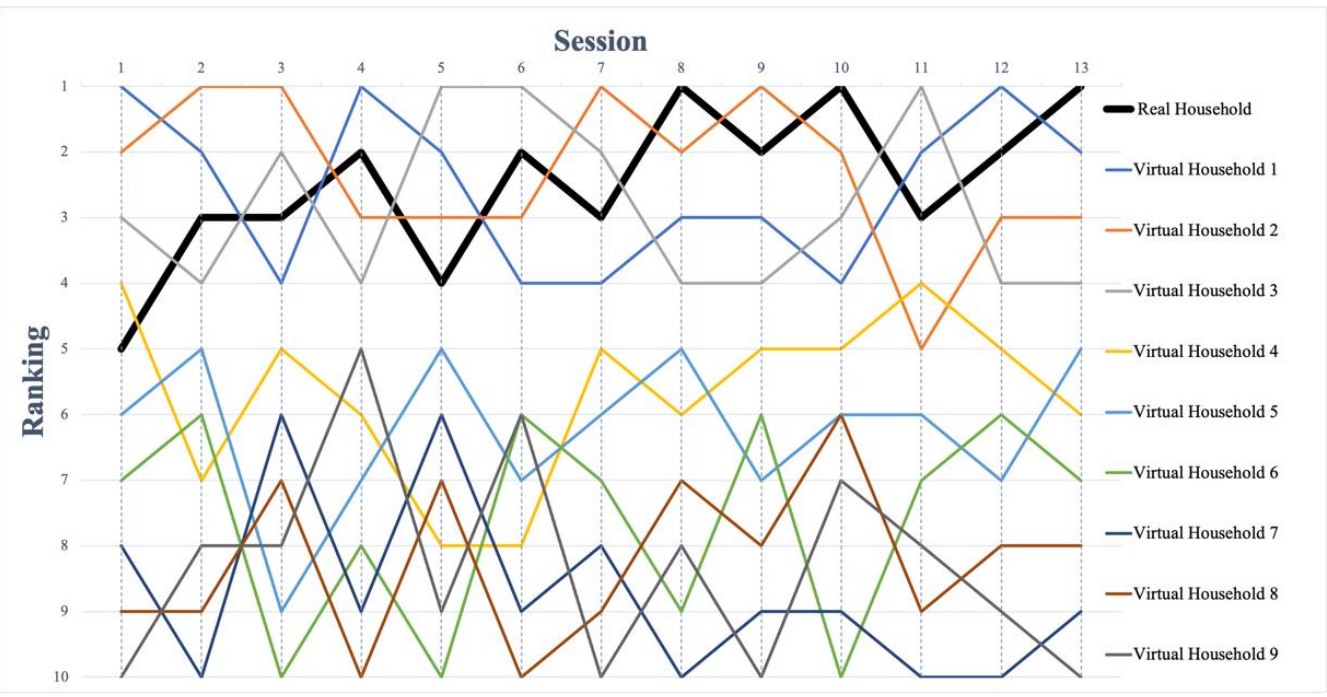

**Figure 9.** Pattern ranking simulated competition.

### 3.3.3. Technical Architecture

Powersaver Game is basically an internet page. The browser of the players' device (tablet, pc and/or laptop) communicates with the webserver. The browser loads the game

from the server, and, during game play, exchanges information with the server. That allows the server to store data such as missions that are completed and started, quiz questions that are answered, current scores, etcetera. From an external server, we retrieve energy data. On our server at the university, we poll several times a day if new energy data are available and, if so, store them on our server. There are three sources that can change the status of the game. First of all, there are user actions: starting a mission, answering questions or completing a mission. Two, there is the delivery of energy data. This is not initiated by the user but is collected by an independent process. The third type of change happens through the passage of time. After the start of a mission, it takes 24 h before the user can complete the mission. In the meantime, the user is supposed to take action to save on energy.

Since the previous study, as described in Fijnheer et al. [3], there has been a change in the acquisition of energy data. This used to be done with the help of the smart home automatization company BeNext and now is done via the service provider EDSN. EDSN develops and operates the Dutch energy data hub on behalf of the Dutch transmission and distribution system operators.

There are technical differences between both organizations in the way the acquisition of energy data is done. The smart energy meter that records energy data in a household can be accessed via two interfaces: Port 1 (P1) and Port 4 (P4). The P1 interface of the smart energy meter is a physical connector, to which a device can be connected to record standings. This is the interface that was used in the old setup (Figure 10, old P1 interface above). The datalogger device was developed by BeNext and connected via the internet to the BeNext servers to store energy data. The P4 interface works more indirectly (Figure 10, new P4 interface below). The energy system operator can read out the hourly usage of gas and electricity via the mobile GPRS-network. All energy system operators in The Netherlands send those data to the service provider EDSN, which allows authorized parties like us to access it. Of course, a household has to give explicit permission for this. So, both interfaces provide usage information. However, with the P4 interface, the data becomes available once a day. Depending on the energy system operator, the data of the previous day arrive between 7 p.m. and 2 a.m. With the P1 interface, it is possible to have a setup such that the data is available almost instantly. For the game play, the latter is clearly preferable. The results in Powersaver Game are based on energy savings during missions. However, with the P4 interface, it may take more than a day to get access to those data. The drawback of using the P1 port is that there needs to be extra hardware installed in the household. A datalogger device, a cable between the smart energy meter and this device and some more hardware to connect it to the internet. This datalogger device has to be delivered and installed, and at the end of the game deinstalled and collected. On top of the hardware purchases, unexpected expenses had to be made to support households to install hardware and return it afterward. Furthermore, our aim was to involve more households than in previous studies. The drawbacks of using the P1 port were considered larger than the advantages. So, we switched to the P4 port for the current study. To get swift gameplay, we take energy data of a day old to compute the score. That means that changes in energy usage influence the score somewhat later than may be expected by the user.

There are several programming languages and libraries used in the system. In the web browser, the main programming language used is Typescript (Version 3.6, Microsoft). Some older parts of the application use JavaScript (ECMAScript 2018, Mozilla). The Phaser.io (Version 2.7, MIT License) game framework is used as a game engine. The server backend uses C# (Version 8.0, Microsoft) as language and ASP.Net (Version 4.6, Microsoft) as the library to communicate with the browser application. There is also a MySQL (Version 5.7, GPL) database on the server to store game data. The communication with EDSN, the provider of energy data, is done with the help of the Java language and libraries via SOAP. This interface is dictated by EDSN. The communication is secured by means of a certificate. The certification is a formal process, which takes some time and effort. The certificate is signed by KPN, a Dutch telecom provider. KPN verifies the identity of the applicant and

verifies that he is the owner of the domain name in the certificate. The certificate is used to encrypt and sign every message to EDSN.

In total, it took us almost one year to get authorization and rebuild the technical architecture.

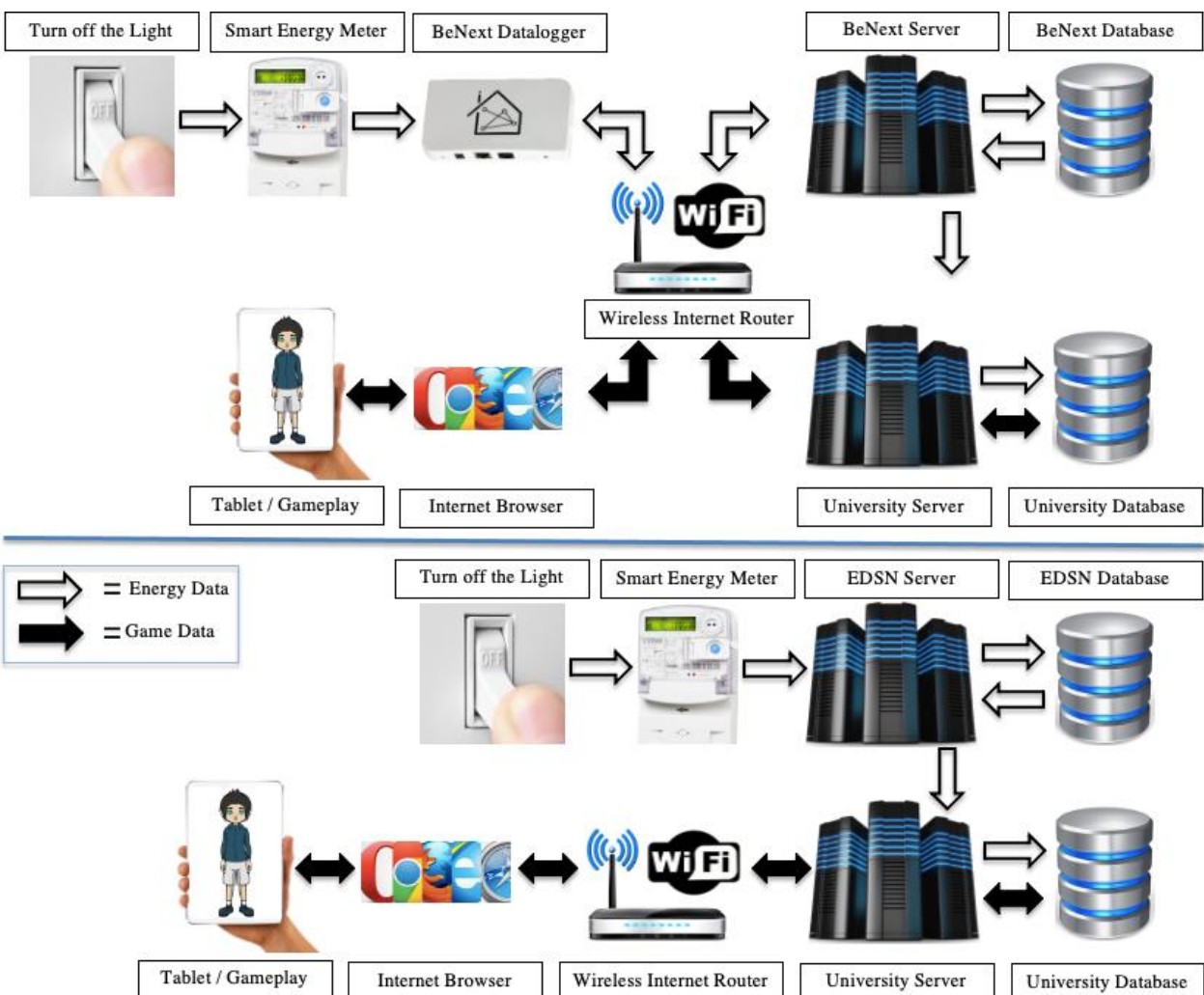

**Figure 10.** Technical architecture Powersaver Game: old P1 interface above, new P4 interface below the line.

### 3.4. Measurements

An overview of measurements is presented in Table 3. Participants completed an online pretest as well as an online posttest questionnaire to assess their attitude towards sustainable energy consumption-related topics and knowledge level towards household energy conservation. For attitude measures, both questionnaires included 30 statements to be rated on a 7-point Likert-scale ranging from strongly disagree to strongly agree. Krosnick and Petty [70] indicate that strength-related attributes of attitudes can be categorized in affective, cognitive and behavior intention components. In our questionnaire, we only used statements from affective and cognitive categories, because behavior intention to conserve energy in the household was already present by voluntary registration to participate in this experiment. Change in behavior intention would be detected when households quit gaming and do not conserve energy. In the pretest and posttest, the same statements on the same topics are used. Of those, 15 statements are regarding micro-level attitude topics (about sustainable energy consumption in a household) as well as 15 statements regarding macro-level attitude topics (10 statements about sustainable energy in general and 5 statements about sustainability). Macro-statements were composed partly based on previous research on attitudes toward sustainability [5], and involve more general and

important aspects of sustainability. With this approach, we measure specific hierarchical attributes, on macro-level and micro-level, of the object of sustainable energy attitude. It can be assumed that micro-level attitude topics are nested within macro-level attitude topics [71]. Beforehand, it was not clear how they influence each other and behaviors change when the intervention takes place.

**Table 3.** Overview of measurements.

| | Pre Intervention | Intervention | Post Intervention |
|---|---|---|---|
| kWh electricity & m³ gas consumption data from the smart energy meter | 21 days monitoring | 39 days (SD 10 days) monitoring | 21 days monitoring |
| Macro and micro attitude assessment | 30 statements *: 15 micro-level attitude and 15 macro-level attitude | | 30 statements *: 15 micro-level attitude and 15 macro-level attitude |
| Knowledge level | 12 Multiple-choice questions including 4-answer options | | 12 multiple-choice questions including 4-answer options |
| Evaluation of application | | | 14 statements *: 11 about learning to save energy with the game and 3 about the competition feature. One open question about suggestions for improvements. |

* 7-Point Likert-scale ranging from strongly disagree to strongly agree.

We used 12 multiple-choice questions including 4 answer options per question for knowledge measures. These multiple-choice questions are related to the content about energy conservation from Powersaver Game. The same multiple-choice questions are used in both the pretest and posttest.

Behavior, in the form of energy consumption, is monitored for 21 days before the intervention to set a good baseline of average energy consumption. Both kWh electricity and m³ gas consumption are monitored from the smart energy meter. In both applications, the user is getting feedback (on energy use and savings) during the intervention. After the intervention, the energy consumption is again monitored for 21 days to examine the impact of the intervention.

To evaluate Powersaver Game itself in the online posttest questionnaire 14 statements, rated on a 7-point Likert-scale ranging from strongly disagree to strongly agree, and 1 open question, are used. Of those, 11 statements are about learning to save energy with the game. These measures were composed based on previous research on evaluation in serious games [72]. The 3 remaining statements are about the competition feature. Two questions are for the participants in the control condition and are about pre-expectations of a competition feature and the estimated effect of missing it. One question about the competition feature is for the participants in the intervention condition and is about the assumed effect of it. The last question of the posttest is an open question where was asked to give suggestions for improvements.

*3.5. Procedure*

The participants in this study were recruited using different methods and communication channels such as social media, digital newsletters and public lectures. We have put most of our effort into sending direct mail. We did this by sending 3000 randomly selected households in a representative Dutch municipality a letter with the request to register for participation. Participants registered at the end of 2019 and the beginning of 2020 using an online form. They could participate when the technical situation of their energy supply (e.g., presence of smart energy meter) was adequate.

In March 2020, 31 participants from 18 households filled in the online pretest. To monitor real energy consumption in this period also the connection of the smart energy meter and game database was made. It took at least 21 days of monitoring to set a firm baseline.

All participants above 12 years replied to the pretest questionnaire about attitude and knowledge measurements. Participants were randomly assigned to conditions; however, we took care that there was a global matching between conditions on the composition of the household (adults and children) and energy consumption beforehand (higher or lower than average of the same type of households in The Netherlands). Knowledge scores and attitude towards energy conservation are not used in this assignment process, because all participants scored very low on knowledge and very high on attitude. All household types are equally represented in each condition; finally, 10 households are randomly assigned to the intervention 'competition' condition and 8 households are assigned to the control 'noncompetition' condition.

The intervention started in March 2020 and ended in May 2020. Some households ended at the beginning of June, due to delays in starting new sessions. During this period, the energy consumption is measured by monitoring the energy consumption data from the smart energy meter. The energy measurement stopped after at least 21 days from the end of the intervention. All 18 households that started finished on schedule. When a household finished all the sessions, they were asked to fill in the online posttest. On average the intervention took 39 days (SD 10 days), and no significant difference between conditions is detected.

## 4. Results

The effects on energy conservation, knowledge measures and attitude measures are presented below. Energy conservation between the competition condition and control 'non-competition' condition is based on 10 households from the competition condition and 8 households from the control condition. Unfortunately, part of the post-measurements fell outside the heating season. In this season, Dutch households warm their houses using gas boilers because the maximum outside temperature is below 18.5 degrees. This causes a large difference in m$^3$ gas consumption between the heating and non-heating seasons. Although the heating season has ended somewhere in the post-measurements period, still, conditions can be compared because the seasonal effect is for both the same.

### 4.1. Energy Conservation Measures

The results in energy conservation, during the 21 days post-intervention period, comparing the 10 households in the competition condition and 8 households in the control condition are presented in Table 4. The average energy consumption per day from 21 days after the intervention is for all households compared to the consumption over 21 days before the intervention. The difference in percentage change of total energy consumption (%$\Delta$ kWh electricity + %$\Delta$ m$^3$ gas/2) is presented as well as, separately, the percentage change in consumption in kWh electricity and m$^3$ gas, respectively. As household conservation of both kWh electricity and m$^3$ gas are related, we conducted first a multivariate analysis of variance (MANOVA). There was no significant difference between both competition and control condition when considered jointly on the variables conservation of total energy, kWh electricity and m$^3$ gas: Wilk's $\Lambda = 0.77$, $F (2, 15) = 2.19$, $p = 0.146$, partial $\eta^2 = 0.226$. Next, we conducted a correlation analysis. There was no significant correlation detected between electricity and gas consumption, $r (18) = -0.1$, $p = 0.69$. Surprisingly, the intervention had somehow a different positive effect on kWh electricity and m$^3$ gas conservation between households. Probably this is caused by the seasonal effect on m$^3$ gas conservation. In view of these results, also a univariate analysis was conducted. Independent-samples $t$-tests on the gain scores are performed to test if differences in percentages of change between the competition and control condition on each of the energy conservation measures are significant.

At the post-intervention period there is a significant difference of 7.9% in total change in energy conservation between both conditions: $t (16) = -1.83$, $p < 0.05$ (one-tailed test): while the competition condition consumes 51.5% less energy than in the total 21 days pre-intervention period, the control condition consumes 43.6% less energy. The difference

between the groups of 4.3% kWh electricity consumption is not significant: $t$ (17) = $-1.1$, $p > 0.05$. When we look specifically at the conservation of m$^3$ gas there is a significant difference of 9.6% between groups: $t$ (16) = $-1.81$, $p < 0.05$ (one-tailed test). The competition condition consumes almost 71.5% less m$^3$ gas than before, while the control condition consumes 61.9% less m$^3$ gas than before.

**Table 4.** Energy conservation: mean changes, standard deviations, t-statistic and significance level of difference.

| Energy Conservation | Competition | | Control | | Difference | | |
| --- | --- | --- | --- | --- | --- | --- | --- |
| | M | SD | M | SD | M | $t$ | $p$ |
| Total | 51.5% | 10.3 | 43.6% | 7.5 | 7.9% | $-1.83$ | <0.05 * |
| kWh electricity | 15.8% | 8.8 | 11.5% | 8.3 | 4.3% | $-1.1$ | ns |
| m$^3$ gas | 71.5% | 11.2 | 61.9% | 11.2 | 9.6% | $-1.81$ | <0.05 * |

* one-tailed. ns—not significant at 0.05 level.

### 4.2. Knowledge Measures

In total, 31 participants filled out all questionnaires for knowledge and attitude measures; 16 from the competition condition and 15 from the control (non-competition) condition. Paired-samples $t$-tests and Independent-Sample $t$-tests are executed to conclude whether differences between the pretest and posttest are significant within and between conditions. The results in knowledge measures of all participants are presented in Table 5.

**Table 5.** Knowledge in the competition and control condition: means, standard deviations, t-statistic and significance levels of difference.

| Knowledge * | Competition | | Control | | Difference | | |
| --- | --- | --- | --- | --- | --- | --- | --- |
| | M | SD | M | SD | M | $t$ | $p$ |
| Difference (Post-Pre) | 2.19 | 2.56 | 1.2 | 2.18 | 0.99 | 1.15 | ns |
| Pretest | 4.06 | 1.57 | 4.60 | 1.5 | $-0.54$ | $-0.97$ | ns |
| Posttest | 6.25 | 1.88 | 5.80 | 2.18 | 0.45 | 0.62 | ns |
| $t$ | 3.42 | | 2.13 | | | | |
| $p$ | <0.05 | | <0.05 | | | | |

* Maximum score = 12. ns—not significant at 0.05 level.

The average score on the knowledge of participants in both conditions increased. Although the average score in the posttest is not high (the maximum score possible is 12 points), knowledge about energy conservation increased significantly. In the competition condition the average score on knowledge increased 2.19 points: $t$ (15) = 3.42, $p < 0.05$. In the control condition, the average score on knowledge only increased 1.2 points: $t$ (14) = 2.13, $p < 0.05$. However, there is no significant difference between the competition and control conditions in gain scores $t$ (29) = 1.15, $p > 0.05$.

### 4.3. Attitude Measures

As the attitude of both micro-level and macro-level are related, we conducted first a multivariate analysis of variance (MANOVA). There was no significant difference between the competition and control condition when considered jointly on the variables (difference scores) Attitude total, Micro-level and Macro-level attitude: Wilk's $\Lambda$ = 0.95, $F$ (3, 27) = 0.44, $p = 0.73$, partial $\eta^2$ = 0.46. Despite these results, it was decided to analyze the univariate effects of variables to examine whether the manipulation of the competition influenced change in attitude. For attitude measures, paired-samples $t$-tests and independent-sample $t$-tests are executed to investigate whether differences between the pretest and posttest are significant within and between conditions. The results are presented in Table 6.

**Table 6.** Attitude in the competition and control condition: means, standard deviations, t-statistic and significance levels of difference.

| Attitude Total * (Micro-Level and Macro-Level) | Competition | | Control | | Difference | | |
|---|---|---|---|---|---|---|---|
| | M | SD | M | SD | M | t | p |
| Difference (Post-Pre) | 0.08 | 0.41 | 0.16 | 0.34 | −0.08 | −0.61 | ns |
| Pretest | 5.71 | 0.49 | 5.33 | 0.57 | 0.38 | 1.99 | <0.05 *** |
| Posttest | 5.79 | 0.42 | 5.49 | 0.60 | 0.30 | 1.60 | ns |
| t | 0.74 | | 1.81 | | | | |
| p | ns | | <0.05 ** | | | | |
| **Micro-Level Attitude *** | Competition | | Control | | Difference | | |
| | M | SD | M | SD | M | t | p |
| Difference (Post-Pre) | 0.15 | 0.64 | 0.29 | 0.55 | −0.14 | −0.69 | ns |
| Pretest | 5.51 | 0.57 | 5.08 | 0.68 | 0.43 | 1.89 | <0.05 ** |
| Posttest | 5.66 | 0.47 | 5.38 | 0.56 | 0.28 | 1.50 | ns |
| t | 0.92 | | 2.08 | | | | |
| p | ns | | <0.05 *** | | | | |
| **Macro-Level Attitude *** | Competition | | Control | | Difference | | |
| | M | SD | M | SD | M | t | p |
| Difference (Post-Pre) | 0.006 | 0.44 | 0.03 | 0.32 | −0.02 | −0.15 | ns |
| Pretest | 5.91 | 0.53 | 5.58 | 0.80 | 0.33 | 1.36 | ns |
| Posttest | 5.91 | 0.54 | 5.60 | 0.77 | 0.31 | 1.30 | ns |
| t | 0.06 | | 0.33 | | | | |
| p | ns | | ns | | | | |

* Maximum score = 7. ** one-tailed test, *** two-tailed test, ns—not significant at 0.05 level.

Attitude scores from all participants are already high from the beginning. There was a small but significant difference of 0.38 points between conditions before the intervention: Attitude total: $t$ (29) = 1.99, $p < 0.05$. On a deeper level this difference is located in Micro-level attitude: $t$ (29) = 1.89, $p < 0.05$ (one-tailed test). In the posttest this difference disappeared: Attitude total: $t$ (29) = −0.61, $p > 0.05$. Attitude scores in the control condition changed significantly: Attitude total control condition: $t$ (14) = 1.81, $p < 0.05$ (one-tailed test). On a deeper level this difference is detected in Micro-level attitude control condition: $t$ (14) = 2.08, $p < 0.05$. Surprisingly, no significant attitude change was detected in the intervention condition. This indicates that probably a ceiling effect occurred in this condition. Regarding Macro-level attitude and Micro-level attitude, we did not find any significant effects of competition in the change of attitude.

*4.4. Chain of Events; Dependencies between Measures*

To explore if the assumed chain of events, which is based on theoretical reasoning, as expressed in Figure 1 occurs, that is, more accessible knowledge (higher awareness) leads to more attitude change which subsequently leads to greater behavior change (energy conservation change), a path analysis technique is used [73]. A path analysis is an extension of regression analysis and used to test the fit of a correlation matrix against, in this case, the assumed chain of events [74]. To perform a path analysis, first, variables that are related to the events in Figure 1 are selected. Next, the path analysis technique is used to explore the directed dependencies of these variables. The emerged path diagram is then compared to the assumed chain of events in behavior change.

The major variables and associated descriptions used in the path model are presented in Table 7. First, accessible knowledge is related to the Knowledge score at the post-intervention. Second, Attitude change regarding Micro attitude is related to post-intervention Micro-level attitude score. Attitude change regarding Macro attitude is related to post-intervention Macro-level attitude score. Third, Behavior change is related to Energy conservation in the 21-day period after the intervention (post-intervention period).

**Table 7.** Variables path model.

| Knowledge | Knowledge Score, at Post-Intervention |
|---|---|
| Attitude | Micro-level attitude score, at post-intervention<br>Macro-level attitude score, at post-intervention |
| Behavior change | Energy conservation, difference between 21-day post-intervention period and 21-day pre-intervention period of kWh electricity and m$^3$ gas together |

Next, a path analysis technique is used to explore the directed dependencies among the selected variables. A path analysis method estimates both the magnitude and significance of relationships between a set of independent variables and the dependent variables, and between the independent variables [73,75]. In this way, it makes all causal assumptions in the model explicit [74]. Thus, an emerged path diagram consists of independent and dependent variables that have direct and indirect effects on each other, and as a result, can express a chain of events. The terms 'independent variable' and 'dependent variable' can be confusing because, in the procedure, which consists of a series of multiple linear regressions, variables fulfill both roles, except our criterion variable 'Energy conservation'. The calculated path coefficients are standardized regression coefficients (Beta's) showing the direct effect of an independent variable on a dependent variable in the path diagram [73,74]. Beta coefficients indicate how much a predictor—if significant—contributes to the performance on the criterion variable. It is noticeable that all independent variables in our path diagram are affected by each other and therefore have an indirect, mediating effect [73] on the dependent variable Energy conservation. In this study, IBM SPSS AMOS 27.0 was used. A chi-square test, also called the likelihood ratio test, was performed to assess the overall fit of the model. A finding of non-significance corresponds to an adequate model [74].

The first path diagram, presented in Figure 11, is a direct path from the variables Macro-level attitude, Micro-level attitude and Knowledge to Energy conservation. The outcome of $\chi^2$ (3, $N$ = 30) =18.594, $p < 0.05$ indicates that this model is not adequate. There is only a significant path from Macro-level attitude to Energy conservation ($\beta$ = 0.339).

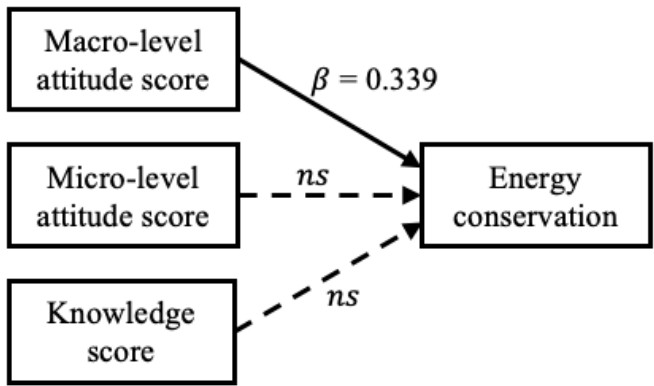

**Figure 11.** First path diagram.

In the second path diagram, presented in Figure 12, the significant path between Macro-level attitude and Energy conservation, as in Figure 11, remains. However, the path runs from the variables Micro-level attitude and Knowledge via Macro-level attitude to Energy conservation. The outcome of $\chi^2$ (3, $N$ = 30) =8.381, $p < 0.05$ indicates that this model is also not adequate. Though, in this path diagram, only the path from Knowledge to Macro-level attitude score is not significant.

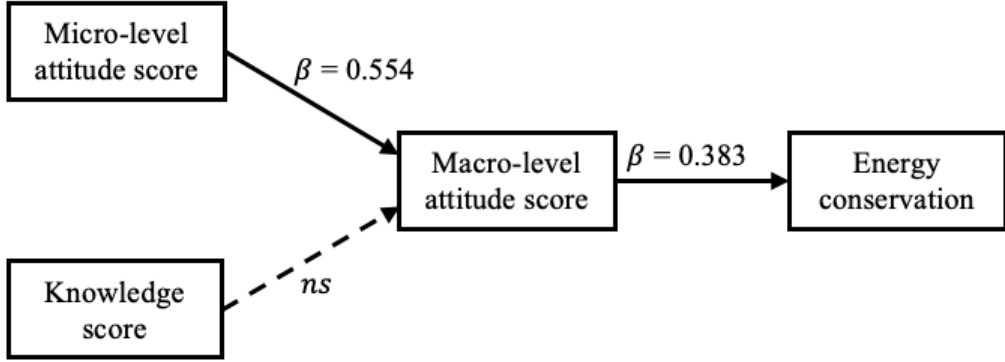

**Figure 12.** Second path diagram.

In the third path diagram, presented in Figure 13, the path runs from the variables Macro-level attitude and Knowledge via Micro-level attitude to Energy conservation. Compared to the second model, Micro-level attitude and Macro-level attitude have therefore switched places. Although this model is adequate, $\chi^2$ (3, $N = 30$) = 5.245, $p = 0.16$, the most important path from Micro-level attitude to the criterion variable Energy conservation is not significant.

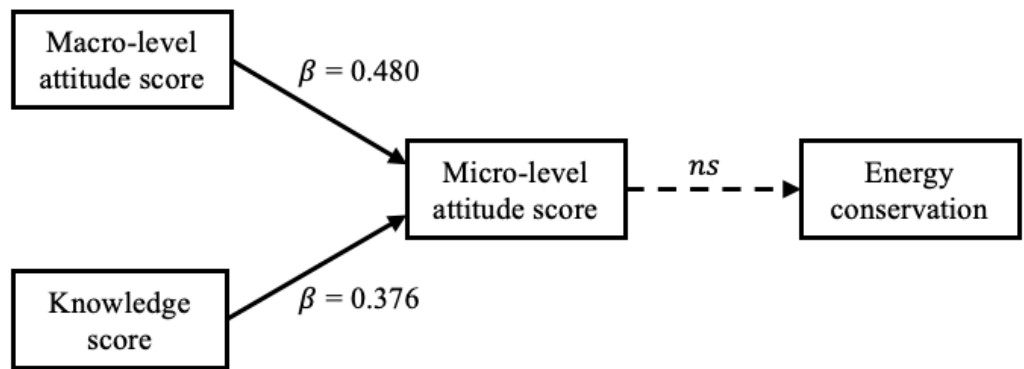

**Figure 13.** Third path diagram.

In the fourth and overall path diagram, presented in Figure 14 (below), the path runs first from the independent variable Knowledge to Micro-level attitude, second from Micro-level attitude to Macro-level attitude and third from Macro-level attitude to the criterion variable Energy conservation. The outcome of $\chi^2$ (3, $N = 30$) = 0.577, $p = 0.90$ indicates that this model is adequate, and all paths are significant.

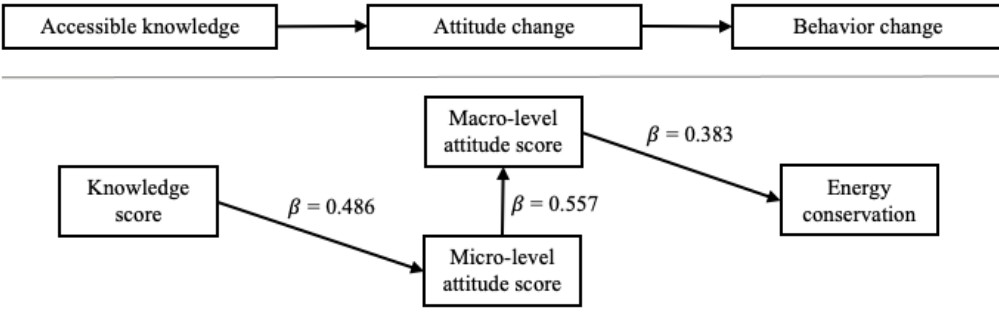

**Figure 14.** Assumed chain of events in behavior change (**above**) and overall path diagram (**below**).

The outcomes in Figure 14 (below the line) generally support the assumed chain of events as expressed in Figure 14 (above the line). The same relationships between variables emerge empirically and indicate the same direction of causality: First, more accessible

knowledge (higher awareness) provided by Powersaver Game, is a significant predictor variable of Micro-level attitude. Secondly, Micro-level attitude is a predictor of Macro-level attitude. Thirdly, Macro-level attitude is a predictor of behavior change as expressed in Energy conservation.

### 4.5. User Evaluation of Powersaver Game

The 31 participants evaluated Powersaver Game by means of 11 statements about learning to save energy with the game and one open question where participants were asked to give suggestions for improvement. The score on learning to save energy with Powersaver Game is on average 4.7 on a 7-point Likert-scale, and the standard deviation is 1.3. There is no significant difference, at the 0.05 level, between groups. The following valuable suggestions for improvement are given; 1. automatic (push-)notifications when a session has ended, 2. shorter mission time (e.g., 20 h), 3. shorter mission texts and 4. general game art improvements (game buttons, animations and saving overviews).

Implementation of adaptation features are also recommended by the participants, specifically pre-selection of appliances and energy-saving activities that are relevant to particular households, and composition and appearance of game avatars.

#### Competition Evaluation

The competition feature is evaluated by using 3 statements, rated on a 7-point Likert-scale ranging from strongly disagree to strongly agree. One statement is for participants in the intervention competition condition and 2 statements are for participants in the control 'non-competition' condition. The participants in the intervention 'competition' condition responded positively, mean 4.4 (SD = 1.7), on the statement that the competition feature stimulates to be more involved in Powersaver Game. The participants in the control 'non-competition' condition scored relatively low, mean 2.8 (SD = 1.4), on the statement if they assumed in advance to compete with other households. Despite this low expectation, they indicate that they would be more motivated, mean 4.1 (SD = 1.8), to accomplish energy conservation missions, if they had to compete with other households in Powersaver Game.

### 4.6. Validity

To ensure validity, we used extensive questionnaires in the pre and post-intervention period for knowledge and attitude measures that are based on previous research, and we monitored actual energy consumption over a long period. Nevertheless, during the pre-intervention period, the Dutch government decided to take COVID-19 measures which influenced the regular energy consumption of households. People had to work at home, if possible, and schools were closed. These COVID-19 measures continued during the intervention and post-intervention period. To ensure the validity of the measures, we decided to analyze data from the pre-intervention before the COVID-19 measures were taken. When energy consumption was lower than in the COVID-19 measures period, these are the days that people are partly not at home, it has been replaced for days further back in time when people are the whole day at home. It was possible to obtain this historical data which was stored in the smart energy meter itself. As a result, the total energy consumption in the pre-intervention period is in line with the situation where people work at home and/or at home instead of school.

Another validity issue is the composition of the households. Eleven of eighteen households consist of families with children. Although we would like to have more of each type of household represented in this study, all household types are equally represented in each condition. A more general constraint is the limited number of households participating in this study, although still significant differences are found.

Finally, the selection of participants is potentially a threat to the validity of the current study. All participants had a high positive attitude towards reduction of energy consumption which on the one hand made it difficult to find changes in attitude, and on the other hand, it restricts the range of participants to which we can generalize.

## 5. Conclusions and Discussion

In previous research, we concluded that 'Powersaver Game', which includes reality by using reality enhanced games principles, is effective in learning people to save energy in the household and to actually do that for the long term [3]. The aim of this 'Reality Enhanced Games' approach is to optimize the transfer between the game world and the real world by feeding a serious game with the user's real-world activities. Players are then immersed in real-life situations that are generated by user interaction with a virtual environment. In the present study, we examine the long-term effects of the persuasive feature social interaction by means of competition on participants' knowledge, attitude and behavior with respect to sustainable energy consumption with Powersaver Game. For our approach, we compare Powersaver Game including a competition feature versus the same game excluding a competition feature.

Firstly, we conclude based on the results of this study that energy consumption changed significantly in the long term. A reality enhanced game, with or without a competition feature, is thus effective in learning people to save energy in the household and to actually do that for the long term. Similar studies also presented positive results but had several shortcomings such as; 1. lack of a control condition, 2. short intervention time, 3. no use of real consumption measurements, 4. implementation of reality could be better, 5. limited number of variables are measured and 5. lack of pre-measurements and post-measurements [2,50], which altogether could explain that the positive effect on energy conservation in our study is higher than in previous studies. What is unique about this research is that we used real-time measurements of $m^3$ gas and kWh electricity consumption. We have therefore not limited ourselves to just knowledge and attitude measurements.

Secondly and most importantly, we found a positive effect of competition on behavior change: the 10 households in the intervention 'competition' condition conserved 8% more energy during the 21 days post-intervention period, compared to the 8 households in the 'non-competition' control condition. And this gain effect was significant. This was mainly caused by $m^3$ gas conservation, while there was no significant difference between both conditions in kWh electricity conservation. Probably this latter difference is not significant because of the high standard deviation in kWh electricity conservation. Despite the fact that there is only a modest (but significant) difference between conditions, participants confirm in post-intervention measures that competition stimulates engagement. Besides energy conservation, no further differences were detected between conditions.

In both conditions knowledge about saving energy at home increased, however, we do not find an effect of introducing competition. It is somehow unexpected that knowledge measures are low in both pre- and post-measurements. We might assume that knowledge transfer from the game to participants progresses in (routine) behavior, but not in increased explicit reproducibility of that knowledge at the test.

The attitude scores on micro-level and macro-level are extremely high and both nearly on the same level. The intervention led only to minor changes and no effect of competition on attitude scores was found. Due to this, a ceiling effect regarding attitude could be the case, resulting in a minor gain in attitude. Though attitude scores were still a significant predictor for energy conservation in the long term as the path analyses showed. Krosnick and Petty [70] argue that the more extreme an attitude is, the more a person likes the object of the attitude and the more likely it is to direct behavior. All participants have a high attitude score, and therefore an extreme attitude towards energy conservation.

The earlier mentioned chain of events of behavior change aligns with the results of the path analysis we performed (see Figure 14). Higher awareness (more accessible knowledge) for a longer period leads to attitude change which in turn results in behavior change in the long term, particularly the macro-attitude plays here a significant role. Also, important to notice is that both knowledge and micro-attitude are playing an indirect role; the influence on energy conservation runs via macro-attitude. So, there is no direct link between micro-attitude and energy conservation, and no direct link between knowledge and energy conservation. We conclude that knowledge about energy conservation that

transfers from the game to the participants is a trigger for attitude about sustainable energy consumption in a household (micro-attitude). Subsequently, attitude about sustainable energy consumption in a household (micro-attitude) acts as a trigger for attitude about sustainable energy in general and sustainability (macro-attitude), which ultimately leads to actual behavior in the form of energy conservation. A practical implication is that energy conservation is stimulated primarily by macro-attitude. So macro-attitude is more important than micro-attitude as demonstrated by the Beta coefficients in the overall path diagram.

Constraints of this study are the start of the COVID-19 measures during the pre-intervention period and ending of the heating season in the post-intervention period, although this occurred equally in both conditions. Another constraint is the limited number of households participating in this study. This limitation unfortunately also occurs in related studies [2,3,50] and points to the difficulties of this kind of research, even with the technologically more user-friendly architecture. It appears that the general public is rather reluctant to participate in this kind of study. It is worthwhile to note that, although the number of households was limited, still significant differences are found. There is a possibility to scale up the number of participants if a large(r) campaign, with the involvement of (semi-) government agencies and energy aggregators, to recruit households is launched.

Despite the long intervention time, participants remained engaged during the whole intervention, because all households finished the game and the overall evaluation is positive. This suggests that the game is effective in stimulating participants long term involvement in household energy conservation activities. Possible modifications for improvements are that the mission's texts and time can be shortened, and more attention can be given to adaptation features and the composition and appearance of game avatars. Adaptive or personalized game elements, such as our competition feature design, could be adjusted to stimulate desired behavior [76].

Future research in reality-enhanced games should continue to focus on the game characteristics that contribute to the game's persuasiveness [11]. To bring the research field on energy reduction games theoretically, but also practically further, it would be useful that in addition to competition the research question "*Which persuasive features of a reality enhanced game exactly promote lasting changes in knowledge, attitude and behavior regarding sustainable energy use of households?*" is also applied to other features. For that purpose, we are planning, in the next phase of research, to continue a 'value added' approach [6]. Here we will examine the effects of three persuasive features—separately and combined—on participants' knowledge, attitude and behavior with respect to sustainable energy consumption with a new extended version of Powersaver Game. First, personal relevance by means of adaptive customized avatars. The avatars will represent the residents of the household, including physical appearance and clothing, and their state will depend on the game results [77–79]. This feature can enhance considerably the engagement of users. Second, explanatory feedback through simulations of future scenarios using machine learning. Adaptive algorithms will improve the effectiveness of future scenarios, through better forecasting and simulations, that are presented [80,81]. More insight will be created when future effects of behavior changes are presented. Furthermore, energy conservation missions will be adjusted to the household characteristics and its performances [82,83]. Third, collaborative-competitiveness by means of competing groups of households while collaborating within groups. As a follow-up to this study, in addition to collaboration within the household, there will also be collaboration between households. Collaboration within teams of households is stimulated, facilitated and monitored by providing in-game communication tools. Competition is then stimulated by forming groups of households based on location (e.g., neighborhoods or city). The benefits of putting together both collaboration and competition are not yet completely clear. However, it is promising, because collaboration seems to have a greater impact on social skills, whereas competition

seems to have greater influence on motivation to spend more effort and concentration on an activity [84]. It is expected that both features will reinforce each other [36].

**Author Contributions:** Conceptualization, J.D.F. and H.v.O.; methodology, J.D.F. and H.v.O.; software, G.-J.G.; validation, J.D.F. and H.v.O.; formal analysis, J.D.F. and H.v.O.; investigation, J.D.F.; resources, J.D.F. and H.v.O.; data curation, J.D.F. and G.-J.G.; writing—original draft preparation, J.D.F., H.v.O. and G.-J.G.; writing—review and editing, J.D.F., H.v.O. and R.C.V.; visualization, J.D.F.; supervision, H.v.O. and R.C.V.; project administration, J.D.F. All authors have read and agreed to the published version of the manuscript.

**Funding:** This research received no external funding.

**Institutional Review Board Statement:** Not applicable.

**Informed Consent Statement:** Informed consent was obtained from all households/participants involved in the study.

**Data Availability Statement:** Not applicable.

**Acknowledgments:** The authors thank all the participants in the study.

**Conflicts of Interest:** The authors declare no conflict of interest.

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
