# Peer review of "Competition in a Household Energy Conservation Game"

_sustainability, doi:10.3390/su132111991_

Round 1

Reviewer 1 Report

Overall, the work is sound, but the presentation is missing important details. See separate file.

Author Response

Dear Sir / Madam,

Thank you very much for your positive and supporting feedback. And thank you for your constructive suggestions for improvements.

Concerning your suggested ‘details that need clarification’ and ‘typos and minor points’, we made some additions, which you can find in the new version of the paper. And below you find our response to each of your suggestion.

  1. How were the participants recruited? This is described in Section 3.5, but is too vague in my opinion. Shouldn’t the choice of recruitment venues be visible to detect any unconsidered bias? Some participant bias is discussed in section 3.4, and that is appreciated. Are these platforms generic, targeted as specific groups, etc?

Author reaction: we have added some text in section 3.5 to clear this up (mainly in the first few sentences).

  1. Please give more example levers that the households have to affect their consumption besides washing clothes on low temperatures, which is mentioned four times. Is this really the only one.

Author reaction: We provided much more examples in general; and give more details in the section 3.3 design

  1. It is suggested that different households have a different have different length intervention periods and different post-intervention periods (see lines 487–492). Details of this should be given by condition.

Author reaction: There is no significant difference between conditions. We have added an extra sentence: “On average the intervention took 39 days (SD 10 days), and no significant difference between conditions is detected.”

  1. There are no details on how the simulator of the other families works. What are the principles behind it? Does the simulator receive inputs from the real households to keep them in a competitive, and therefore motivated position? The importance of this is discussed in section 3.3.1, but how it is done and how the simulation works is not discussed that I could see.

Author reaction: The principles are mentioned in the first two paragraphs of section 3.3.1 such as: team-based approach, stimulating collaboration and high scores, and equal opponents. We also mention that energy conservation data of virtual households follow a logical pattern based on the real time energy conservation of the real household. And why we choose this approach to stimulate behavior.

  1. There appears to be a (minor) ethical issue in this work. The households in the competitive branch of the experiment were playing against a simulation, but “assumed to play against 9 real households”. It is not clear whether the participants were given misinformation, or merely given no information. Give a justification that this deception is ethical, such as the ethical procedure/committee the project went through, or an argument. If it is the case that the participants were given no reason to believe that the simulation was of real families, how do you know what they assumed?

Author reaction: The participants in the competition condition were not aware of playing against virtual households. The presentation in the game suggests that they are playing against real households. After the intervention, we asked some questions about the effect of the competition feature (see 4.5.1), and received no comments from participants that there was any doubt.

  1. Could the pre and post test questions be made available in an appendix or via an web link. Allowing the community to see the questions in a survey is standard.

Author reaction: The questions will be published and thereby made available in the thesis I’m (Jan Dirk Fijnheer) working on right now. This will be published in Spring 2022. At the moment the questions are only in the Dutch language, and have to be translated.

Typos and minor points

  1. Line 226 missing “on” : Author reaction: Thanks, we implemented your suggestions.
  2. Line 694 “learning people”, teaching people? Author reaction: we are convinced that in this context learning is a better word than teaching.
  3. The first time DGBL is used, indicate what it is by writing it in full with the acronym in parentheses, e.g. Digital Game-Based Learning (DGBL). This could be in section heading 2.2, or in the text the first time it appears. The hyphen needs to be there. Author reaction: The term ‘Digital Game-Based Learning’ and acronym is first used in line 115 in section 2.1. We are convinced it is not appropriate to use the acronym in a/the title (-of 2.2.).
  4. The phrase value added has many meanings and is used in an unusual way here. It seems to mean some kind of feature sensitivity. It is defined on line 257, but is used three times before that. Could it be defined before it is first used? Author reaction: We have made some changes in line 71, where the term is first used, to make clear what it is, already from the introduction: “For that purpose, we have in the next phase of research, applied a ’value added’ approach. Studies using this approach focus on the question which features of a game promote learning. Conditions including selected features are compared to a condition with a base version of the game (Mayer, 2011).”

We hope these changes are sufficient for you.

Kind regards,

Jan Dirk Fijnheer

Reviewer 2 Report

The paper is generally well written. However, I would like to see a separate section "Threats to Validity" explaining all the external factors that may have influenced the result. Some of them have been discussed  and addressed within the paper. But a separate section consolidating all of them would be good. In particular, the influence of COVID, the family structure of partcipants and timing (for example,  families containing school kids may spend more time at home during vacation and hence consume more energy). All such threats should be clearly stated.

Author Response

Dear Sir / Madam,

Thank you very much for your positive and supporting feedback. We have followed your advice by adding a new paragraph: 4.6 “Validity”. Therefore, we removed the text about COVID to this new section and added other validity issues.

We hope this is sufficient for you.

Kind regards,

Jan Dirk Fijnheer

Reviewer 3 Report

Current topic is actual and interesting. By this moment, there is only few scientific research in this area, so every publication is valuable. It is worth continuing this research. Despite the fact the authors conducted complicated tests, they managed to present them in an understandable method in the article.

Author Response

Dear Sir / Madam,

Thank you very much for your positive and supporting feedback. We also agree that this topic is very actual and interesting. We hope that more researchers from several disciplines will be involved in similar research, and that our publication gives them support and inspiration. At the moment we are working on new research funding to continue our research.

Kind regards,

Jan Dirk Fijnheer